# Green and Effective Preparation of α-Hydroxyphosphonates by Ecocatalysis

**DOI:** 10.3390/molecules27103075

**Published:** 2022-05-11

**Authors:** Pola Cybulska, Yves-Marie Legrand, Alicja Babst-Kostecka, Sébastien Diliberto, Anna Leśniewicz, Erwan Oliviero, Valérie Bert, Clotilde Boulanger, Claude Grison, Tomasz K. Olszewski

**Affiliations:** 1Department of Physical and Quantum Chemistry, Faculty of Chemistry, Wrocław University of Science and Technology, Wybrzeże Wyspiańskiego 27, 50-370 Wrocław, Poland; pola.cybulska@pwr.edu.pl; 2Bio-Inspired Chemistry and Ecological Innovations (ChimEco), UMR 5021 CNRS, University of Montpellier, Cap Delta, 1682 rue de la Valsière, 34790 Grabels, France; yves-marie.legrand@umontpellier.fr; 3Department of Environmental Science, The University of Arizona, Tucson, AZ 85721, USA; ababstkostecka@arizona.edu; 4Institut Jean Lamour, UMR 7198 CNRS, University of Lorraine, 57000 Metz, France; sebastien.diliberto@univ-lorraine.fr (S.D.); clotilde.boulanger@univ-lorraine.fr (C.B.); 5Analytical Chemistry and Chemical Metallurgy Division, Faculty of Chemistry, Wrocław University of Science and Technology, Wybrzeże Wyspiańskiego 27, 50-370 Wrocław, Poland; anna.lesniewicz@pwr.edu.pl; 6ICGM, University of Montpellier, CNRS, 34090 Montpellier, France; erwan.oliviero@umontpellier.fr; 7Clean Technologies and Circular Economy Unit, SIT Department, INERIS, Parc Technologique Alata BP 2, 60550 Verneuil en Halatte, France; valerie.bert@ineris.fr

**Keywords:** hydroxyphosphonates, hydrophosphonylation, biomass, green chemistry, aldehydes

## Abstract

A green and effective approach for the synthesis of structurally diversed α-hydroxyphosphonates via hydrophosphonylation of aldehydes under solventless conditions and promoted by biosourced catalysts, called ecocatalysts “Eco-MgZnOx” is presented. Ecocatalysts were prepared from Zn-hyperaccumulating plant species *Arabidopsis halleri*, with simple and benign thermal treatment of leaves rich in Zn, and without any further chemical treatment. The elemental composition and structure of Eco-MgZnOx were characterized by MP–AES, XRPD, HRTEM, and STEM–EDX techniques. These analyses revealed a natural richness in two unusual and valuable mixed zinc–magnesium and iron–magnesium oxides. The ecocatalysts were employed in this study to demonstrate their potential use in hydrophosphonylation of aldehydes, leading to various α-hydroxyphosphonate derivatives, which are critical building blocks in the modern chemical industry. Computational chemistry was performed to help discriminate the role of some of the constituents of the mixed oxide ecocatalysts. High conversions, broad substrate scope, mild reaction conditions, and easy purification of the final products together with simplicity of the preparation of the ecocatalysts are the major advantages of the presented protocol. Additionally, Eco-MgZnOx-P could be recovered and reused for up to five times.

## 1. Introduction

Because of their unique properties, organophosphorus compounds have found many interesting applications in important areas of chemical industry. These applications span from the use of organophosphorus compounds in the preparation of utility chemicals such as flame retardants [1] and anticorrosive coatings and adhesives [2,3,4], through applications as ligands for catalysis [5,6,7,8,9], and finally their use as biologically active compounds, e.g., agrochemicals (insecticides, herbicides, and fungicides), and pharmaceutically active ingredients [10,11,12]. Among the known organophosphorus compounds, the α-hydroxyphosphonic acids, their esters (hydroxyphosphonates), and close derivatives represent an interesting class of molecules endowed with a wide range of properties. They are especially useful as biologically active compounds such as antibiotics (e.g., Valinophos **I** [13], and Fosfazinomycin A **III** [14]); enzyme inhibitors (e.g., renin inhibitors **II** [15], for potential treatment of hypertension and congestive heart failure or neuraminidase inhibitor [16], for treatment of influenza infection, commonly called “the flu”); antiparasitics (e.g., inhibitors of *Plasmodium falciparum* growth **IV** [17], useful in treatment of malaria that is caused by this parasite); and insecticides (e.g., trichlorfon **V**) (Figure 1). Additionally, α-hydroxyphosphonic acids and their esters are very useful scaffolds in organic synthesis often used to prepare more complex molecules [18], e.g., after *O*-allylation and subsequent RCM or isomerization/Claisen rearrangement [19], reaction with primary amines leading to α-aminophosphonates and phosphonic acids [20], phospha-Brook rearrangement [21], reduction [22], halogenation [23], and oxidation leading to ketophosphonates (Figure 1) [24].

Because of the importance of the α-hydroxyphosphonic acids and their esters, the organic chemistry community has a constantly increasing interest for the synthesis of these compounds [18]. The most commonly used methodology to prepare α-hydroxyphosphonates is based on the addition of “P-nucleophiles” (*H*-phosphonates (Pudovik reaction) [25] or trialkyl- or triarylphosphites (Abramov reaction) [26]) to C=O bond in carbonyl compounds. The generally accepted catalytic cycle for the addition of *H*-phosphonates to C=O bond relies on the use of catalysts or reagents either for the activation of the carbonyl compound or the activation of the phosphorus nucleophile (by formation of a phosphanion-phosphite form) or the simultaneous activation of both substrates (Figure 1).

Currently, considering the growing environmental concerns, the development of new environmentally sustainable, safe, and preferably catalytic and solventless protocols leading to α-hydroxyphosphonates is especially desirable [27]. To that end, several attempts to prepare α-hydroxyphosphonates via phosphonylation of carbonyl substrates under solventless conditions and in the presence of various catalysts were reported in the literature. So far, catalysts activating the carbonyl are quite rarely presented in the literature and heteropolyacids of structure H_6_P_2_W_18_O_62_·14H_2_O [28], CeCl_3_·7H_2_O [29], or ammonium metavanadate (NH4VO3) are worth mentioning [30]. More commonly described in the literature are catalysts activating the P-nucleophiles, and for this the cyclopentadienyl ruthenium(II) complex ([RuClCp(PPh_3_)_2_] [31], hydrotalcite MG7 [32], *n*-BuLi [33], amidate ytterbium amide {Yb[N(SiMe_3_)_2_](κ^2^-L^1^)_2_(THF) (L^1^ = C_6_H_5_C(O)NC_6_H_3_ (*^i^*Pr)_2_)} [34], potassium phosphate (K_3_PO_4_) [35], 2-*tert*-butyl-imino-2-diethylamino-1,3-dimethylperhydro-1,3,2-diazapho-sphorine supported on polystyrene (PSsBEMP) [36], methylene-linked pyrrolyl samarium and yttrium amido complexes [37], or molybdenum dichloride dioxide (MoO_2_Cl_2_) [38] are of special importance. Finally, bifunctional catalysts activating both carbonyl and phosphonate substrates are known and here Fe_3_O_4_@SiO_2_-Met-Cu(II) in the presence of *tert*-butyl hydroperoxide (TBHP) [39], Bi(NO_3_)_3_.H_2_O [40], choline chloride [41], bimetallic samarium bis(cyclopentadienyl) derivatives supported by bridged bis(guanidinate) ligands {(CH_3_C_5_H_4_)_2_Sm[(*^i^*PrN)_2_CN(CH_2_)_2_]}_2_ [42], sodium-modified fluoroapatite (Na@FAP) [43], nafion resin-supported oxovanadium(IV) catalyst [44], and sodium-modified-hydroxyapatite (Na-HAP) [45] should be pointed out.

Unfortunately, most of these catalysts are complex molecules in many cases based on the use of metals including not easily available lanthanides and transitions metals, and the use of hazardous ligands, reagents, and metallic salts. It is important to also mention that simple inorganic salts such as CaO [46], MgO [47], or Na_2_CO_3_ [48], are reported to promote the hydrophosphonylation of carbonyl substrates under solventless conditions. However, they are often used in stochiometric amounts; the reactions are substrate specific and, most importantly for green chemistry considerations, the production of the salts is far from environmentally friendly, with harsh production processes, poor impact assessment, and positive carbon footprint. Additionally, examples of the use of specific techniques such as microwave irradiation [49] and biocatalysis [50] as green methodologies leading to hydroxyphosphonates are also described in the literature. However, the use of microwave irradiation is exclusively limited to the use of few substituted aromatic aldehydes, and no examples for the use of aliphatic or heteroaromatic aldehydes, or ketones were reported [49]. Biocatalytic methods [50] can be applied to a wider range of aldehydes, albeit the yields of final products are low (11–40% and 68% yield in the best case) and additionally isolation of pure products requires tedious chromatographic purification.

Despite the abovementioned advances in the preparation of α-hydroxyphosphonates under green conditions, the development of new synthetic protocols toward those important compounds remains desirable. Especially important is the development of efficient, simple, inexpensive, and biosourced catalysts for this transformation.

Recently, a new concept in catalysis termed “ecocatalysis” was introduced by Grison [51,52,53]. This innovative methodology is based on the use of biomass derived from metal-hyperaccumulating plants, used in the phytoextraction of metal contaminated sites, for the preparation of polymetallic catalysts suitable for applications in organic synthesis. This concept was successfully applied for important transformations such as the Lewis and/or Brønsted acid-catalyzed reactions (Diels–Alder reaction, Friedel–Crafts reaction, multicomponent reactions, cascade reactions) [54,55,56,57,58]; coupling reactions (Suzuki, Heck, Sonogashira and Ulmann type reactions) [59,60,61,62,63,64]; reductions (aminoreduction, selective reduction of αβ-unsaturated carbonyl compounds, nitro- and halogeno arenes) [65,66,67]; oxidations (epoxidations, oxidative cleavage, oxidations of alcohols) [68,69,70]; and tandem reactions (tandem carbonyl-ene cyclization, synthesis of substituted pyridines and oxidative iodination of ketones) [71,72,73]. However, examples on the use of ecocatalysis for C-P bond formation have not been reported.

In this work, as the source of biomass, we selected the abundant metal-hyperaccumulating plant, *Arabidopsis halleri*, growing on metal-contaminated slag heaps on post-mining sites in central and western Europe, which has been previously applied in phytoextraction of metal contaminated sites [74,75]. *Arabidopsis halleri* (L.) O’Kane and Al-Shehbaz (family Brassicaceae), referred hereafter as *A. halleri*, is considered one of the most prominent plant model species for its ability to tolerate and hyperaccumulate extremely high concentrations of zinc (Zn) and cadmium (Cd) in its shoots [76,77]. For Zn and Cd, the thresholds for hyperaccumulation have been defined as 3000 and 100 mg kg^−1^, respectively [78]. Thus, these unique traits of *A. halleri* have a potential to be applied in the development of urgently needed new phytomanagement strategies, especially where metal removal from contaminated soils or recovery from plant biomass is desired. In this study, we specifically investigated locations for which extremely high concentrations of Zn both in soil and in plant shoots have previously been reported [79,80].

Herein, we report for the first time on the preparation and the characterization of Eco-MgZnOx ecocatalysts, obtained after simple controlled thermal treatment of the biomass rich in metals. Hydrophosphonylation reaction conditions for a model compound were then explored using the novel ecocatalysts and compared with commercial oxides (pure and mixtures). Next, recyclability of ecocalysts after the formation of α-hydroxyphosphonates under green and sustainable conditions was evaluated. Subsequently, structurally diverse carbonyl substrates have been assessed to verify the range of applicability for the developed methodology. Finally, a theoretical evaluation of the presence of mixed oxides was also performed to guide us in the respective reactivity of each constituent of the mixed ecocatalysts.

## 2. Results and Discussion

### 2.1. Preparation and Characterization of Eco-MgZnOx

#### 2.1.1. Preparation of Eco-MgZnOx

In this study we have used biomass from plant *A. halleri* growing in two different locations in Europe. The first biomass was derived from heavily industrialized region of Poland in the vicinity of the Zn smelter of the Bolesław Mine and Metallurgical Plant near the city of Olkusz [81]. The second biomass was derived from a heavily polluted soil in a former industrial site located in Auby, France (Bois des Asturies) [82]. The idea of the contribution of *A. halleri* was strongly inspired by the fact that this plant has the potential to be applied in the development of new phytomanagement strategies based on metal removal from contaminated soils or recovery from plant biomass. Finding application for this metal-enriched biomass resulting from phytoextraction is therefore highly desirable. Both biomass samples were simply treated under air flow at 550 °C and the resulting thermal residues were directly used as catalysts termed Eco-MgZnOx-P (derived from Polish *A. halleri*) and Eco-MgZnOx-F (derived from French *A. halleri*). The elemental composition of both catalysts was determined by MP–AES and characterized by MP–AES, XRPD, high resolution transmission electron microscopy (HRTEM), scanning transmission electron microscopy (STEM), and energy dispersive X-ray spectroscopy (STEM–EDX) mapping.

#### 2.1.2. Characterization of Eco-MgZnOx by MP–AES Analysis

Table 1 shows the mineral composition of Eco-MgZnOx catalysts obtained with *A. halleri* from France (Eco-MgZnOx-F) and Poland (Eco-MgZnOx-P). The applied thermal treatment under air flow leads to a high loss of organic matter (90% to 95% of mass loss) and to a high concentration of mineral residues. Zn concentrations are high and are almost the same in Eco-MgZnOx-F from France (9.89 wt %) and Eco-MgZnOx-P from Poland (11.44 wt %), which confirms the ability of this species to hyperaccumulate Zn. It should be noted that the average Cd concentration remains below 0.2 wt % for both locations. This observation is particularly interesting, considering that Zn-hyperaccumulating plants are often able to coaccumulate Cd in elevated concentrations; however, toxicity is problematic. Mineral compositions of Eco-MgZnOx-F and Eco-MgZnOx-P catalysts are very similar, except for the magnesium content. Eco-MgZnOx-P exhibits twice the amount of Mg when compared to Eco-MgZnOx-F (respectively, 5.24 wt % and 2.64 wt %).

#### 2.1.3. Characterization of Eco-MgZnOx by X-ray Powder Diffraction

The X-ray powder diffraction (XRPD) analysis of Eco-MgZnOx was performed to identify metallic species under crystalline form. The XRPD diffractograms of Eco-MgZnOx-P and Eco-MgZnOx-F (see Appendix A) highlighted the complexity of the samples with a large number of phases (9 in total) as expected for biomass. Both Eco-MgZnOx-F and Eco-MgZnOx-P catalysts clearly exhibited the presence of CaCO_3_ and K_2_SO_4_. Very interestingly, a Mg_0.88_Zn_0.12_O phase was also observed in both cases and particularly clearly in Eco-MgZnOx-P. This compound is an isotype of periclase MgO, but the lattice parameter is significantly larger. The diffractogram allowed for determining a parameter equal to 4.232 Å instead of 4.213 Å, which is the lattice parameter of pure MgO. This modification of the structure can easily be explained by the substitution of Mg atoms by Zn atoms, which have a larger atomic radius. The formation of magnesium oxide would be very surprising at 550 °C considering that its standard formation temperature is above 800 °C. Interestingly, the presence of additional metallic species (Zn, Fe) could enhance the reactivity of MgO, as suggested in the literature [82,83], and as discussed in the theoretical section. Diffractograms showed the expected presence of ZnO, K_2_ZnSiO_4_, and ZnSiO_3_, which are zinc silicates and used as antireflection coatings [84].

#### 2.1.4. Characterization of Eco-MgZnOx by Electron Microscopy

High resolution transmission electron microscopy (HRTEM) images of Eco-MgZnOx are shown in Appendix A. For both Eco-MgZnOx-P (a) and Eco-MgZnOx-F (b), a high concentration of particles is observed, with two different particle populations mixed together: large particles with sizes ranging from 100 to 600 nm, and small spherical particles with diameter ranging from 10 to 50 nm. Most of the small particles are nested into the larger ones. Scanning transmission electron microscopy and energy dispersive X-ray spectroscopy (STEM–EDX) analysis was used to determine the chemical composition of the Eco-MgZnOx. Figure 2 clearly shows that the larger particles in both Eco-MgZnOx are composed of Ca and O, i.e., calcite. Additionally, at a much lower content, a calcite base in the smaller particles is clearly visible for Eco-MgZnOx-F (Figure 2b), where the small particles are separated from the larger ones. As further evidenced by the superposition images small particles are mainly composed of Mg, Zn, and O, and, to a lesser extent, Fe, which is consistent with MP–AES analysis (Table 1), where Fe wt % is very low. These observations can validate the presence of MgO·ZnO and/or MgO·FeO, as evidenced by XRPD. As a trace element, images of iron are poor in contrast. Electron microscopy thus confirms that the Eco-MgZnOx are composed of large calcite particles mixed with smaller mixed magnesium-based oxides, MgO·ZnO and MgO·FeO particles.

### 2.2. Reactivity of Eco-MgZnOx in the Hydrophosphonylation Reaction

#### 2.2.1. Catalyst Impact on the Formation of Model α-Hydroxyphosphonate **3****a**

We initiated our study for effective preparation of diethyl-(1-hydroxyphenylmethyl) phosphonate (**3a**) via ecocatalysis by reacting benzaldehyde (**1**) with diethyl *H*-phosphonate (**2**), as a model substrate (Table 2), in the presence of Eco-MgZnOx-P (derived from Polish *A. halleri*) and Eco-MgZnOx-F (derived from French *A. halleri*), after optimization of the reaction conditions (see Appendix A). The activity of the ecocatalysts was compared to those of commercial oxides commonly used as catalysts, pure and in mixtures. Owing to the polymetallic composition of the Eco-MgZnOx catalysts, we chose to focus on Zn, Mg, Fe, K, and Ca as the elements present in the catalysts that are the most likely contributors to the catalytic activity.

Based on the reaction mechanism generally accepted for the hydrophosphonylation reaction (Figure 1), the Eco-MgZnOx could be considered as plurifunctional catalyst. Firstly, the presence of Mg species with basic character should be responsible for activation of the *H*-phosphonate and formation of a phosphanion (phosphite form). Secondly, the presence of Zn species with Lewis acid character could be responsible for the activation of the carbonyl carbon making it more electrophilic and prone to reaction with phosphorus nucleophile. The simultaneous activation of both substrates can also be envisaged. Initially, reaction was performed under solventless conditions, and this resulted in the best choice. The reaction proceeded readily at room temperature but with poor to moderate conversions depending on the reaction time (28% and 80% after 1 h and 24 h, respectively). When heating was applied, conversions were greatly improved and reaction time was limited to only 3 h. Further, the amount of Eco-MgZnOx was optimized to the amount Mg (7.0 mol%), Ca (13.6 mol%), and Zn (5.7 mol%) for Eco-MgZnOx-P, and Mg (3.5 mol%), Ca (11.4 mol%), and Zn (4.9 mol%) for Eco-MgZnOx-F (Table 2, entry 2 and 3). Lower amounts of Mg, Ca, and Zn resulted in significant decrease in the reaction conversion (see Appendix A). Importantly, reactions without the plant-based catalysts did not produce any desired α-hydroxyphosphonate (Table 2, entry 1). As a comparison, model reactions were performed in the presence of commercial ZnO, MgO, and their mixtures (Table 2, entries 4–8), and FeO (Table 2, entries 7–8), K_2_CO_3_ (Table 2, entry 9) and CaCO_3_ (Table 2, entry 10) all in the amounts corresponding to their presence in Eco-MgZnOx catalysts. No reaction was observed in the presence of CaCO_3_. In turn, the reaction catalyzed by ZnO led to only to 14% conversion (Table 2, entry 4) whereas for MgO, conversions were higher (70% and 62% for the amount corresponding to Eco-MgZnOx-P and Eco-MgZnOx-F, respectively). The combination of ZnO and MgO gave higher conversion than both oxides separately (Table 2, entry 8), but the obtained values were still lower than the ones obtained with the use of Eco-MgZnOx catalysts. Additionally, use of FeO in combination with MgO and ZnO did not result in a significant increase in the reaction yield. The same was observed for the use of K_2_CO_3_ (Table 2, entry 9). These results clearly show that the catalytic activity of both plant-based catalysts Eco-MgZnOx-P and Eco-MgZnOx-F lies in their polymetallic composition and in the original species present in their plant-based structure.

#### 2.2.2. Recyclability and Reuse of Eco-MgZnOx Catalysts

The possibility of simple and efficient recovery and reuse of the catalyst is a very important aspect of catalysis and green chemistry; therefore, we performed these experiments for Eco-MgZnOx-P and Eco-MgZnOx-F. For this purpose, hydrophosphonylation of benzaldehyde (**1**) with diethyl *H*-phosphonate (**2**) under optimized conditions was used as model reaction. After each run, ethyl acetate was added to the reaction mixture and Eco-MgZnOx-P or Eco-MgZnOx-F was easily separated by centrifugation and washed with an additional portion of ethyl acetate. The catalysts were dried at 500 °C for 5 h and then used directly in the next reaction. During these tests we observed a clear difference between reactions catalyzed by Eco-MgZnOx-P and Eco-MgZnOx-F (Table 3). Clearly, in the case of the Eco-MgZnO-F, loss of activity was observed in the second reaction (Table 3). In turn, the Eco-MgZnO-P remained active and could be used 5 times without significant loss of activity (88% conversion after 5th run). The MP–AES analysis of the mineral composition of the catalysts recovered after each reaction showed that the amount of Zn, Mg, and Ca was decreasing; therefore, the activity of both catalysts could indeed be associated with the concentration of these elements (see Appendix A). In both cases (Eco-MgZnOx-P and Eco-MgZnOx-F), during work-up, a small quantity of ecocatalyst could not be recovered for the following reaction, as shown by MP–AES. This could be explained by the leaching of some ecocatalyst with the formed product, hydroxyphonates, known to be good chelating agents. The main difference between Eco-MgZnOx-P and Eco-MgZnOx-F stands in the initial concentration of Mg, which is less than half in the latter ecocatalyst. The reactivity loss in the case of Eco-MgZnOx-F recycling is probably due to the fact that the minimum ecocatalyst threshold is reached, for the reaction to proceed fully, as early as the 2nd recycling step. Based on all of these results, it became clear that Zn is not the only key element responsible for the high catalytic activity of Eco-MgZnOx catalysts. Instead, the presence of Mg was found to have significant influence on the outcome of the studied reaction. Indeed, the amount of Mg present in Eco-MgZnOx-F (derived from French *A. halleri*) was lower than in the case of Eco-MgZnOx-P (derived from Polish *A. halleri*) (3.5 mol% versus 7.0 mol%, respectively), and this influenced the conversions of the reactions after recovery and reuse of the Eco-MgZnOx. The loss of Mg during each run is responsible for the loss of catalytic activity and since the Eco-MgZnOx-P contains more Mg, it can be used 5 times; for the Eco-MgZnOx-F, the conversion falls below 60% as soon as the second run.

#### 2.2.3. Scope of the Hydrophosphonylation Reaction of Carbonyl Substrates Catalyzed by Eco-MgZnOx

After showing that the Eco-MgZnOx are highly efficient and recyclable on model hydrophosphonylation reaction leading to compound **3a**, we examined the scope of the reaction with respect to various carbonyl substrates and *H*-phosphonate substrates, as shown in Figure 2 and Figure 3, respectively. High conversions (up to 99%) were observed for reactions of various aromatic, aliphatic, and heteroaromatic carbonyl compounds **1a–m** with model diethyl *H*-phosphonate **2** and expected α-hydroxyphosphonate **3a–m** were isolated with good yields (up to 90%), as summarized in Figure 2. Initially, benzaldehyde and its derivatives with electron-withdrawing substituents such as 4-Cl or 4-NO_2_ were used as substrates and gave the expected α-hydroxyphosphonates **3a–c** in good, isolated yields (82%, 80%, and 72%, respectively). Similarly, benzaldehyde derivatives with electron-donating substituents such as 4-Me or sterically demanding 2,4,6-trimethylbenzaldehyde were found to react easily to form the desired products **3d** and **3e** (82% and 75% isolated yields, respectively). Additionally, aliphatic aldehydes worked smoothly under the optimized reaction conditions and gave the expected α-hydroxyphosphonates **3f–i** in good, isolated yields (up to 84%). Similarly, heteroaromatic aldehydes were well tolerated in the reaction and afforded products **3j–l** with good, isolated yields (up to 87%) (Figure 2). An interesting example is the compound **3m**, derivative of cinnamaldehyde, which was obtained selectively as a product of addition of *H*-phosphonate to the carbonyl group and not to the double C=C bond (competitive reaction). Despite extensive investigations, however, we found that ketones were unsuitable for the hydrophosponylation reaction catalyzed by Eco-MgZnOx. The best observed conversion for acetophenone was in the range of 35–37% and reaction required 20 h at 70 °C (Figure 2). Changing the ketone to aliphatic or heteroaromatic derivative did not improve the reaction outcome. Subsequently, we examined the scope of the reaction with respect to various *H*-phosphonates (Figure 3). Using model carbonyl substrates such as benzaldehyde (**1a**), isovaleraldehyde (**1f**), and 3-pyridinecarboxyaldehyde (**1j**), we tested structurally diverse *H*-phosphonates such as alkyl derived dimethyl-, dibutyl-, and diisopropyl *H*-phosphonates, along with aromatic dibenzyl *H*-phosphonate (Figure 3). The desired α-hydroxyphosphonates **5a–h** were formed with high conversions (up to 99%) and isolated yields (up to 92%) for dimethyl and dibenzyl *H*-phosphonates albeit, with moderate conversions (up to 70%) and isolated yields (62%) for long-chain dibutyl- and sterically crowded diisopropyl *H*-phosphonates. This could be explained by lower reactivity and additional steric hindrance present by the two latter *H*-phosphonates. In all cases (Figure 2 and Figure 3) after reaction completion, ethyl acetate was added to the reaction mixture and Eco-MgZnOx catalysts were easily separated by centrifugation and washed with additional portion of ethyl acetate. The organic layer was concentrated on vacuum and products were further purified. For aromatic and heteroaromatic α-hydroxyphosphonates, which were solids, crystallization from a mixture of diethyl ether/hexane was used. In the case of oily aliphatic α-hydroxyphosphonate, preparative TLC was applied to obtain analytically pure samples. Isolated products were fully characterized by standard spectroscopic techniques (see Appendix A for more details).

Finally, we also performed scaling-up of the hydrophosphonylation reaction to 4.6 mmol, 5 times larger scale than for experiments in Figure 2 and Figure 3 with Eco-MgZnOx-F, and the pure desired product **3a** was obtained without problems after simple crystallization with 88% yield (1.0 g).

### 2.3. Theoretical Assessment of the Catalytic Activity of Eco-MgZnOx in Hydrophosphonylation Reaction

To obtain additional information on the ecocatalyzed phosphonylation reaction and rationalize its possible mechanism, theoretical calculation was performed. It must be mentioned that Eco-MgZnOx are not conventional catalysts with well-defined and well-known composition and structure. Therefore, the impact of the presence of one or another element in the ecocatalytic matrix is so far difficult to establish. The HP(O)(OMe)_2_ was selected as model *H*-phosphonate and its reactivity with several metal oxide mixtures was assessed ((Mg_1.0_Zn_0.0_)O, (Mg_0.9_Zn_0.1_)O and (Mg_0.9_Fe_0.1_)O). Thus, modeling the reactivity of inorganic bulk species is not a trivial task and the relevance of using ionic/cluster models instead of periodic model and standard hybrid functionals instead of complex models was thoroughly considered. Indeed, as Xu et al. explained as early as 1999, a bulk solid can be regarded as the sum of fragments and cluster models of metal oxides can be built with three principles, namely, a neutrality principle, a stoichiometry principle, and a coordination principle [85]. Woodley et al. proposed recently models of magnesium and calcium oxides and showed thermodynamically stable structures (global and local minima) for nanoclusters resembling rock-salt phases [86]. Senthilvelan et al. [87] produced several Zn_x_O_y_ model clusters and did a comparison between experimental and theoretical results in order to improve molecular modeling. They investigated the electronic structure of guanine interacting with different-sized ZnO clusters (Zn_2_O_2_, Zn_3_O_3_, and Zn_4_O_4_) and identified a stable Guanine–Zn_2_O_2_ cluster by DFT theory. Meanwhile, Fabris et al. [88], examined how accurate hybrid functionals (B3LYP, PBE0) are compared to high-level quantum chemistry methods such as coupled cluster (CC) quantum chemistry calculations when studying catalyzed reactions by metal oxide (CoO). They found that these functionals lead to fair agreement with the computationally expensive CC calculations. The stability of cubane units was shown by X-ray absorption fine structure (EXAFS) spectroscopy [89,90] and confirmed by theoretical models [91]. The same authors suggested that the ion model, even simpler than various cubane units, mimics the main features of the active site and is therefore a reasonable system for benchmarking. Related to hydroxyl deprotonation and by using both experiments and theoretical tools, Guesmi et al. [92] showed how surface hydroxyls enhance MgO reactivity in basic catalysis. Indeed, the basicity of the surface induces a deprotonation of the hydroxyl group, followed, in their case, by C-C bond breaking and proton transfer. Based on all these studies combining experimental and theoretical results, we proposed to use the PBE0 hybrid functionals to model the reaction pathways of previously presented systems with simple yet robust models of metal oxides, as demonstrated earlier [88]. The MgO cluster of 5 × 2 × 2 was constructed to model the oxide surface. Energy of intermediates was evaluated to identify the most likely reaction pathways. Furthermore, considering that the ethyl group in HP(O)(OEt)_2_, held by the phosphite derivative, does not significantly change the reactivity and that it increases the calculation times, the whole theoretical study was performed using the dimethyl- instead of the diethyl-*H*-phosphonate HP(O)(OMe)_2_. As shown on Figure 1, the equilibrium between the *H*-phosphonate and phosphanion (phosphite form) is well described [18]. In the presence of a basic surface such as a MgO surface (Lewis base), the P-OH moiety can easily be deprotonated. To test this first step in the reaction, the model HP(O)(OMe)_2_ is placed in presence of the model MgO cluster. Upon the geometry optimization, a proton transfer is observed from the phosphite to the MgO surface; more specifically, a Mg-O bond breaks, a hydroxyl is formed, and an ionic complex between the anionic remaining oxygen and the dangling cationic Mg site is observed (Figure 3), as described in detail by Petitjean et al. [92]. This reaction can take place on various locations of the MgO cluster, but the most stable form is measured when the phosphite reacts on the tip of the cluster (Figure 3) with a stabilization energy of 81.8 kcal mol^−1^. The phosphite was also placed in presence of other forms of mixed MgO cluster, i.e., (Mg_0.9_Fe_0.1_)O and (Mg_0.9_Zn_0.1_)O, for the same reaction. No significant difference in terms of energy was observed for this proton transfer step (see Appendix A). This confirms the assumptions that the initial deprotonation step is enhanced by the presence of a basic MgO surface, but not particularly by other elements present in the Eco-MgZnOx species, such as iron or zinc.

The following step in the catalytic reaction cycle involves the attack of benzaldehyde carbonyl by the nucleophilic phosphorus atom. To assess this step, the stable geometry between the metal clusters and the phosphite was appended to the benzaldehyde and this constituted the new starting geometry. To promote the possible stabilization of the benzaldehyde carbonyl by the metallic cluster, the molecule was favorably oriented to reduce calculation time. The geometry optimization revealed significant difference while the initial geometries were identical. The only case where the spontaneous attack of P on the carbonyl occurs is for the (Mg_0.9_Zn_0.1_)O. The stabilization energy is again significant (36.1 kcal mol^−1^). In the case of pure MgO cluster and MgO-Fe enriched cluster, the stable geometry is a nonbonded complex between the cluster and the benzaldehyde, but no P-C bond formation can be observed. This is a fair indication that the (Mg_0.9_Zn_0.1_)O is more efficient in activating the aldehyde and which favors the phosphonylation reaction. The final step of the overall catalytic cycle is the proton transfer from metallic cluster to the new organic species and the release the final product. A slight amount of energy is required for this step, which is likely provided from the heat of reaction. This evaluation confirms that the reaction probably occurs as expected (deprotonation, nucleophilic attack, proton transfer and release), and that the zinc in the (Mg_0.9_Zn_0.1_)O cluster can be responsible for most of the yield improvement seen experimentally, compared to magnesium oxide (Table 2).

## 3. Materials and Methods

### 3.1. General Information

Melting points were determined at constant 5 °C/min with Digimelt Apparatus using the standard open capillary method and are uncorrected. ^1^H, ^13^C, and ^31^P NMR spectra were collected on a Jeol 400 yh instrument (400 MHz for ^1^H NMR, 162 MHz for ^31^P NMR, and 100 MHz for ^13^C NMR). NMR spectra recorded in CDCl_3_ were referenced to the respective residual ^1^H or ^13^C signals of the solvents. The reported *J* values are those observed from the splitting patterns in the spectrum and may not reflect the true coupling constant values. High-resolution mass spectra were collected using electrospray ionization on a Waters LCT Premier XE TOF instrument. Preparative TLC was carried out using silica gel 60 precoated plates.

### 3.2. Plant Growth and Biomass Information—Study Sites and Sampling of Plant Material

The sampling included the metal-contaminated locations of *A. halleri* in southern Poland and northern France.One of the coauthors, V. Bert, identified and collected the plant. The plant was identified for the first time in 1944 at Auby, France, 20 km from Belgium, by a botanist, A. Berton. Since then, the site became an international reference as several metallicolous plants were found, including the Zn- and Cd-hyperaccumulator Arabidopsis halleri ((formerly *Cardaminopsis halleri* (L.) Hayek)). Numerous specimens of *A. halleri* have been deposited in herbariums in France, including some specimens from Poland (ReColNat voucher: P06489474) and Belgium (ReColNat voucher: P06803999), and we could confirm the accurate identification of the plant owing to a clear match. The Polish location represents a low elevation site located in one of the most heavily industrialized regions of Poland, the vicinity of the Zn smelter of the Bolesław Mine and Metallurgical Plant near the city of Olkusz [93]. Accordingly, total soil Zn, Cd, and Pb concentrations at this sampling location are as much as 3911 ± 340, 1045 ± 115, 28 ± 3 mg kg^−1^, respectively [79]. In France, *A. halleri* plants were collected in a former industrial soil contaminated with Zn and Cd (7000 and 60 mg kg^−1^, respectively) [73]. *A. halleri* rosettes were collected from each location in early August 2019.

### 3.3. Preparation of the Eco-MgZnOx

Harvested biomass was washed twice in deionized water, air dried in an oven at 60 °C for 12 h under atmospheric pressure, and then ground. The powder obtained was thermally treated in an oven under air flow at 550 °C for 6 h.

### 3.4. Characterization of the Eco-MgZnOx

X-ray diffraction (XRPD) data measurements on the samples dried at 110 °C for 2 h were performed by using a BRUKER diffractometer (D8 advance, with a Cu Kα radiation λ = 1.54086 Å) equipped with a LynxEye detector.

Mineral composition of the catalysts was determined using an Agilent 4200 microwave plasma and atomic emission spectrometer (MP–AES) coupled with a SPS4 autosampler. The samples were digested in 6 mL of reversed aqua regia (1:2 hydrochloric acid (37%): nitric acid (65%)) under an Anton Paar Multiwave Go microwave-assisted digestion, with the following program: 20 °C to 164 °C in 20 min, then 10 min isothermal at 164 °C. Samples were filtered and then diluted to 0.2 g L^−1^ in nitric acid (1%). Three blanks were recorded for each step of the dilution procedure. Three analyses were carried out for each sample to determine the standard deviation of the measurement.

Sample preparation for electron microscopy by ultramicrotomy: powder obtained after preparation of the catalysts was embedded in LR White resin (aromatic acrylic resin) and cut with an ultramicrotome Leica UC7 (Plateforme MEA, University of Montpellier). Ultrathin sections (70 nm) were collected on 300-mesh carbon-coated copper grids. Electron microscopy: HRTEM and STEM–EDX were performed on a JEOL 2200 FS operated at 200 kV (field emission gun). HRTEM images were recorded with a Gatan Ultrascan 4000SP CCD camera, 4096 × 4096 pixels. EDX spectra were acquired on a SDD Oxford Instrument XMaxN 100 TLE (100 mm^2^—windowless).

### 3.5. General Procedure for Hydrophosphonylation Reaction with Eco-MgZnOx Catalysts

Appropriate aldehyde (0.92 mmol), H-phosphonate (0.92 mmol), and appropriate Eco-MgZnOx catalyst (Eco-MgZnOx-P (5.7 mol% of Zn) or Eco-MgZnOx-P (4.9 mol% of Zn)) were placed in a glass vial and stirred (at 50 °C for aromatic aldehydes and at 70 °C for aliphatic and heteroaromatic aldehydes) for 3 h. After that time, ethyl acetate was added (2 mL) and the reaction mixture was centrifuged (6000 rpm/5 min). The organic layer was separated, and an additional portion of ethyl acetate (2 mL) was added to the remaining catalyst and the centrifugation repeated (this process of washing of the catalyst was repeated 4 times). The combined organic layers were evaporated under vacuum and the remaining crude product was purified by crystallization or preparative TLC.

### 3.6. General Recovery and Reuse of Eco-MgZnOx Catalysts

Appropriate aldehyde (0.92 mmol), H-phosphonate (0.92 mmol), and appropriate Eco-MgZnOx catalyst (Eco-MgZnOx-P (5.7 mol% of Zn) or Eco-MgZnOx-F (4.9 mol% of Zn)) were placed in a glass vial and stirred at 50 °C (for aromatic aldehydes and at 70 °C for aliphatic and heteroaromatic aldehydes) for 3 h. After that time, ethyl acetate was added (2 mL) and the reaction mixture was centrifuged (6000 rpm/5 min). The organic layer was separated, and an additional portion of ethyl acetate (2 mL) was added to the remaining catalyst and the centrifugation repeated (this process of washing of the catalyst was repeated 4 times). The combined organic layers were evaporated under vacuum and the remaining crude product was purified by crystallization or preparative TLC. The remaining solid catalyst was dried in an oven (500 °C for 5 h) and then used for the next reaction. The process could be repeated up to 5 times in the case of Eco-MgZnOx-P.

### 3.7. Experimental Theoretical Section

Density functional theory (DFT) calculations were performed for geometries and energies as implemented by Gaussian 16 C.01 [94]. GaussView 6.0 [95] was used for visualization of orbitals. The SCF convergence default was used for Gaussian and the symmetry constraint was ignored. A hybrid functional, which includes a mixture of Hartree–Fock exchange with DFT exchange–correlation, was used. The PBE1PBE functional that uses 25% exchange and 75% correlation weighting is known in the literature as PBE0. The 1996 pure functional of Perdew, Burke, and Ernzerhof was made into a hybrid by Adamo [96]. The double -ζ basis set def2-SVP [97] with polarization functions was used. The D3 version of Grimme dispersion with Becke–Johnson damping was added for empirical dispersion [98], typically used for noncovalent interactions, supramolecular complexes in solution.

## 4. Conclusions

In this study, a new generation of ecocatalysts (Eco-MgZnOx) was easily prepared from Zn-hyperaccumulating plant species, which are used in mining restoration and phytomanagement. A controlled thermal treatment of Zn-rich plant biomass led directly to green basic catalysts, Eco-MgZnOx, thus avoiding any further chemical activation. Characterization by MP–AES, XRPD, HRTEM and STEM–EDX revealed an original polymetallic system. The Eco-MgZnOx ecocatalysts prepared from two origins of the *A. halleri* plant species demonstrated an excellent catalytic potential in the hydroxyphosphonylation, in short reaction times (3–5 h), at moderate temperatures (50–70 °C), without solvent nor activation phase [99]. A theoretical assessment allowed for evaluation of the role of some of the constituents of the mixed oxide ecocatalysts, and particularly the added value of the MgZnO mixture. A wide range of hydroxyphosphonates were efficiently synthesized in high yields from structurally different aldehydes and *H*-phosphonates. Moreover, Eco-MgZnOx-P can be recycled and reused for up to five runs. The Eco-MgZnOx catalysts open a new perspective for the valorization of Zn-rich plant biomass through a green and sustainable synthesis of hydroxyphosphonates. This new generation of ecocatalysts exhibits environmental and scientific benefits.

## Data Availability

Not applicable.

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
