# Peer review of "Green and Effective Preparation of α-Hydroxyphosphonates by Ecocatalysis"

_molecules, 2022, doi:10.3390/molecules27103075_

Round 1
Reviewer 1 Report
Please find below my review of the manuscript 1705289
This manuscript reports the preparation of ecocatalysts from Zn-hyperaccumulating plant species Arabidopsis halleri from two sites (France and Poland). The elemental composition and structure of the 2 ecocatalysts were investigated. The ecocatalysts were used to carry out hydrophosphonylation of aldehydes. In addition, the recyclability and reuse of the Eco-MgZnOx catalysts was studies. Finally, computational chemistry was performed to establish the role of some of the constituents of the mixed oxide ecocatalysts.
The article is clearly laid out. All the key elements abstract, introduction, materials and methods, results and discussion, and conclusion are present. The title describes clearly the article.
Introduction section
The introduction summarizes well previous relevant research. Here are few suggestions:
Line 83 replace “2-tert-butyl-imino-2-diethylamino-1,3-” with “2-tert-butyl-imino-2-diethylamino-1,3-”
Line 129-130 replace “Arabidopsis halleri (L.) O'Kane and Al-Shehbaz” by “Arabidopsis halleri (L.) O'Kane and Al-Shehbaz”
Results and Discussion section
Line 161 replace “550 oC“ with “550 °C“
Line 279 replace “500 oC” with “500 °C“
Scheme 3 title
Line 354 replace “aReaction conditions” by “aReaction conditions”
Line 356 replace “50oC” by “50 °C“
Line 357 replace “31P NMR” by “31P NMR”
Materials and Methods Section
Section 3.1 General information: except the first sentence which is little different, the rest of the section is also found in the supplementary section. Is it necessary to write this general information twice (in the main manuscript and in the supporting information file)?
Section 3.2: Who has identified the plant in order to make sure that the plant was well identified? Was a voucher specimen deposited in an herbarium in France and in Poland? What is(are) the deposit number(s)? This information should be included in this section to confirm the right identification of the plant.
Line 465 replace “grinded” by “ground”
Line 490 replace “50 oC (for aromatic aldehydes and at 70 oC” by “50 °C (for aromatic aldehydes and at 70 °C”
Line 500 replace “70 oC” by “70 °C”
Line 507 replace “500 oC” by “500 °C”
References section
Some titles are not written uniformly (some have words inside the title written with capital letters and others do not). Please check.
Lines 614 and 615: do you have the titles?
Write the name of the journals in the abbreviated form for references 21, 31, 39, 42, 48, 51, 55-57, 61, 64, 65, 68, 73-75, 77-78, 80-83, 94.
Line 620 replace “H6P2W18O62•14H2O” by “H6P2W18O62•14H2O”
Line 621 replace “CeCl3.7H2O” by “CeCl3.7H2O”
Line 625 replace “αhydroxyphosphonates” by “α-hydroxyphosphonates”
Line 630 replace “n-BuLi” by “n-BuLi”
Line 631 replace “Org. Lett. 2014, 16,” by “Org. Lett. 2014, 16,”
Line 643 replace “MoO2Cl2” by “[MoO2Cl2]”
Line 645 replace “Fe3O4@SiO2” by “Fe3O4@SiO2”
Line 647 replace “Bi(NO3)3.5H2O” by ““Bi(NO3)3.5H2O”
Line 668: “15478” is not written in the same font as the rest of the reference
Line 672 write Molecules in italics
Lines 737, 745, 748, 751-752, 754: write Arabidopsis halleri in italics
Line 739-740 replace the title “Do Arabidopsis halleri from nonmetallicolous populations accumulate zinc and cadmium more effectively than those from metallicolous populations” by the title “Zinc tolerance and accumulation in metallicolous and nonmetallicolous populations of Arabidopsis halleri (Brassicaceae)” which corresponds to the authors' name, year, journal, volume and page of reference 77.
Line 744: replace “Dietrich, C.C.; Tandy, S.; Banaś, A.; Murawska, K.; Korzeniak, U.; Łopata, B.; Babst-Kostecka, A.” by “Dietrich, C.C.; Tandy, S.; Murawska-Wlodarczyk, K.; Banaś, A.; Korzeniak, U.; Seget, B.; Babst-Kostecka, A.”
Line 747 replace “Neilsona,” by “Neilson,”
Line 759 replace “ZnSiO3ZnSiO3 and Zn2SiO4” by “ZnSiO3ZnSiO3 and Zn2SiO4”
Supporting Information
Page 4 Diethyl (2,4,6-trimethyl-1-hydroxybenzyl)phosphonate (3e): replace “129.7 68..7 (d, J=158,3 Hz),“ by “129.7, 68.7 (d, J=158.3 Hz),“ and replace “C14H23O4PNa [M+Na]+” by “C14H23O4PNa [M+Na]+”
Page 5 Diethyl 2-ethyl-1-hydroxybutylphosphonate (3i) replace “16.5 (d, J=5.7 Hz),” by “16.5 (d, J=5.7 Hz),”
Page 5 Diethyl 1-hydroxy(pyridine-3-yl)methylphosphonate (3j) : replace “148.2 (d, J=6.2Hz),” by “148.2 (d, J=6.2Hz),”
Page 6 Diethyl (1-hydroxy-3-phenyl)prop-2-enyl-phosphonate (3m): check the section “70.3, 133.4, 123.6, 69.4 (d, J=160.9 Hz),” 133.4 and 123.6 should not be between 70.3 and 69.4. Please check.
Page 6 Dimethyl (1-hydroxyphenylmethyl)phosphonate (5b): replace “54.1 (d, J= 7,4Hz), 53.7 (d, J= 7,4Hz);” by “54.1 (d, J= 7.4Hz), 53.7 (d, J= 7.4Hz);”
Page 7 Dibenzyl (1-hydroxyphenylmethyl)phosphonate (5d) : replace “5.09-4.83-(m, 5H),” by “5.09-4.83 (m, 5H),” and complete “68. (m);” 68.8 ?
Page 7 Dimethyl 1-hydroxy-3-methyl-butylphosphonate (5e): replace “65.8 (d, J=160.6 Hz), 53.3-53.5(m), 39.9, 24.1 (d, J=14.3 Hz),” by “65.8 (d, J=160.6 Hz), 53.3-53.5(m), 39.9, 24.1 (d, J=14.3 Hz),”
Page 7 Dibenzyl 1-hydroxy-3-methyl-butylphosphonate (5f) replace “68.1 (t, J=6.8 Hz),” by “68.1 (t, J=6.8 Hz),”
Page 7 Dimethyl 1-hydroxy(pyridine-3-yl)methylphosphonate (5g): replace “δ 23.2 HRMS“ by ” δ 23.2; HRMS”
Page 8 line 3 replace “Rection proceeded already at room temperature“ by “Reaction proceeded already at room temperature “
Page 8 lines 4, 5, 8, 19, 21, 23 and 24: replace “Table S1” by “Table SI1
For the figure at the bottom of the page 10: could you please replace 16,21 by 16.21, 4,32 by 4.32, 14,77 by 14.77, 1,69 by 1.69, 7,22 by 7.22, 15,92 by 15.92, 3,72 by 3.72, 6,96 by 6.96, 14,46 by 14.46, 1,26 by 1.26, 5,23 by 5.23, 13,65 by 13.65, 2,94 by 2.94, 5,27 by 5.27, 15,42 by 15.42, 0,79 by 0.79, 3,45 by 3.45.
This paper is appropriate for publication in Molecules. The manuscript is well written in Standard American English. The referee recommends the publication of this article in Molecules after all the revisions are performed.
Author Response
We would like to thank you for giving us the final opportunity to correct our manuscript. We believe that we have addressed the referee’s comments and have attached the necessary changes below.
Reviewer 1.
This manuscript reports the preparation of ecocatalysts from Zn-hyperaccumulating plant species Arabidopsis halleri from two sites (France and Poland). The elemental composition and structure of the 2 ecocatalysts were investigated. The ecocatalysts were used to carry out hydrophosphonylation of aldehydes. In addition, the recyclability and reuse of the Eco-MgZnOx catalysts was studies. Finally, computational chemistry was performed to establish the role of some of the constituents of the mixed oxide ecocatalysts.
The article is clearly laid out. All the key elements abstract, introduction, materials and methods, results and discussion, and conclusion are present. The title describes clearly the article.
Introduction section
The introduction summarizes well previous relevant research. Here are few suggestions:
Line 83 replace “2-tert-butyl-imino-2-diethylamino-1,3-” with “2-tert-butyl-imino-2-diethylamino-1,3-”
Answer: We thank the Reviewer for that remark. This was corrected.
Line 129-130 replace “Arabidopsis halleri (L.) O'Kane and Al-Shehbaz” by “Arabidopsis halleri (L.) O'Kane and Al-Shehbaz”
Answer: We thank the Reviewer for that remark. This was corrected.
Results and Discussion section
Line 161 replace “550 oC“ with “550 °C“
Answer: We thank the Reviewer for that remark. This was corrected.
Line 279 replace “500 oC” with “500 °C“
Answer: We thank the Reviewer for that remark. This was corrected
Scheme 3 title
Line 354 replace “aReaction conditions” by “aReaction conditions”
Answer: We thank the Reviewer for that remark. This was corrected.
Line 356 replace “50oC” by “50 °C“
Answer: We thank the Reviewer for that remark. This was corrected.
Line 357 replace “31P NMR” by “31P NMR”
Answer: We thank the Reviewer for that remark. This was corrected.
Materials and Methods Section
Section 3.1 General information: except the first sentence which is little different, the rest of the section is also found in the supplementary section. Is it necessary to write this general information twice (in the main manuscript and in the supporting information file)?
Answer: We thank the Reviewer for that remark. This was corrected by removing the text from supporting information file.
Section 3.2: Who has identified the plant in order to make sure that the plant was well identified? Was a voucher specimen deposited in an herbarium in France and in Poland? What is(are) the deposit number(s)? This information should be included in this section to confirm the right identification of the plant.
Answer: We thank the Reviewer for that remark. One of the co-author, V. Bert, has identified the plant and collected it in Auby (France). There is no doubt about the identification of the plant on this site. The plant was identified for the first time at Auby (France, 20 km from Belgium) by a botanist, A. Berton, in 1944 (A. Berton (1946) Présentation de plantes ArabisHalleri, Armeriaelongata, OEnanthefluviatilis, Galinsogaparvifloradiscoidea, Bulletin de la Société Botanique de France, 93:5-6, 139-145, DOI: 10.1080/00378941.1946.10834513). Since then, the site became an international reference as several metallicolous plants, of which the Zn and Cd hyperaccumulator Arabidopsis halleri (formerly Cardaminopsis halleri (L.) Hayek), were found. Many studies have been performed on the plant model A. halleri collected on this site (e.g. Huguet et al. 2012; Gomez-Balderas et al. 2014; Deyris et al. 2018; Grignet et al. 2021). Our team has more than a decade of experience with A. halleri, both in field and lab studies, including also genetic identification of the here investigated populations (e.g. refs 79,80,82). We are very confident that we identified the plant material correctly.
Numerous specimens of A. halleri have been deposited in herbariums in France, including some specimens from Poland (ReColNat voucher: P06489474) and Belgium (close to collection site, ReColNat voucher: P06803999.
Line 465 replace “grinded” by “ground”
Answer: We thank the Reviewer for that remark. This was corrected.
Line 490 replace “50 oC (for aromatic aldehydes and at 70 oC” by “50 °C (for aromatic aldehydes and at 70 °C”
Line 500 replace “70 oC” by “70 °C”
Answer: We thank the Reviewer for that remark. This was corrected.
Line 507 replace “500 oC” by “500 °C”
Answer: We thank the Reviewer for that remark. This was corrected.
References section
Some titles are not written uniformly (some have words inside the title written with capital letters and others do not). Please check.
Answer: We thank the Reviewer for that remark. This was corrected.
Lines 614 and 615: do you have the titles?
Answer: We thank the Reviewer for that remark. The titles were included.
Write the name of the journals in the abbreviated form for references 21, 31, 39, 42, 48, 51, 55-57, 61, 64, 65, 68, 73-75, 77-78, 80-83, 94.
Answer: We thank the Reviewer for that remark. This was corrected as requested
Line 620 replace “H6P2W18O62•14H2O” by “H6P2W18O62•14H2O”
Answer: We thank the Reviewer for that remark. This was corrected as requested
Line 621 replace “CeCl3.7H2O” by “CeCl3.7H2O”
Answer: We thank the Reviewer for that remark. This was corrected as requested
Line 625 replace “αhydroxyphosphonates” by “α-hydroxyphosphonates”
Answer: We thank the Reviewer for that remark. This was corrected as requested
Line 630 replace “n-BuLi” by “n-BuLi”
Answer: We thank the Reviewer for that remark. This was corrected as requested
Line 631 replace “Org. Lett. 2014, 16,” by “Org. Lett. 2014, 16,”
Answer: We thank the Reviewer for that remark. This was corrected as requested
Line 643 replace “MoO2Cl2” by “[MoO2Cl2]”
Answer: We thank the Reviewer for that remark. This was corrected as requested
Line 645 replace “Fe3O4@SiO2” by “Fe3O4@SiO2”
Answer: We thank the Reviewer for that remark. This was corrected as requested
Line 647 replace “Bi(NO3)3.5H2O” by ““Bi(NO3)3.5H2O”
Answer: We thank the Reviewer for that remark. This was corrected as requested
Line 668: “15478” is not written in the same font as the rest of the reference
Answer: We thank the Reviewer for that remark. This was corrected as requested
Line 672 write Molecules in italics
Answer: We thank the Reviewer for that remark. This was corrected as requested
Lines 737, 745, 748, 751-752, 754: write Arabidopsis halleri in italics
Answer: We thank the Reviewer for that remark. This was corrected as requested
Line 739-740 replace the title “Do Arabidopsis halleri from nonmetallicolous populations accumulate zinc and cadmium more effectively than those from metallicolous populations” by the title “Zinc tolerance and accumulation in metallicolous and nonmetallicolous populations of Arabidopsis halleri (Brassicaceae)” which corresponds to the authors' name, year, journal, volume and page of reference 77.
Answer: We thank the Reviewer for that remark. This was corrected as requested.
Line 744: replace “Dietrich, C.C.; Tandy, S.; Banaś, A.; Murawska, K.; Korzeniak, U.; Łopata, B.; Babst-Kostecka, A.” by “Dietrich, C.C.; Tandy, S.; Murawska-Wlodarczyk, K.; Banaś, A.; Korzeniak, U.; Seget, B.; Babst-Kostecka, A.”
Answer: We thank the Reviewer for that remark. This was corrected as requested
Line 747 replace “Neilsona,” by “Neilson,”
Answer: We thank the Reviewer for that remark. This was corrected as requested
Line 759 replace “ZnSiO3ZnSiO3 and Zn2SiO4” by “ZnSiO3ZnSiO3 and Zn2SiO4”
Answer: We thank the Reviewer for that remark. This was corrected as requested
Supporting Information
Page 4 Diethyl (2,4,6-trimethyl-1-hydroxybenzyl)phosphonate (3e): replace “129.7 68..7 (d, J=158,3 Hz),“ by “129.7, 68.7 (d, J=158.3 Hz),“ and replace “C14H23O4PNa [M+Na]+” by “C14H23O4PNa [M+Na]+”
Answer: We thank the Reviewer for that remark. This was corrected as requested.
Page 5 Diethyl 2-ethyl-1-hydroxybutylphosphonate (3i) replace “16.5 (d, J=5.7 Hz),” by “16.5 (d, J=5.7 Hz),”
Answer: We thank the Reviewer for that remark. This was corrected as requested.
Page 5 Diethyl 1-hydroxy(pyridine-3-yl)methylphosphonate (3j) : replace “148.2 (d, J=6.2Hz),” by “148.2 (d, J=6.2Hz),”
Answer: We thank the Reviewer for that remark. This was corrected as requested.
Page 6 Diethyl (1-hydroxy-3-phenyl)prop-2-enyl-phosphonate (3m): check the section “70.3, 133.4, 123.6, 69.4 (d, J=160.9 Hz),” 133.4 and 123.6 should not be between 70.3 and 69.4. Please check.
Answer: We thank the Reviewer for that remark. This was corrected.
Page 6 Dimethyl (1-hydroxyphenylmethyl)phosphonate (5b): replace “54.1 (d, J= 7,4Hz), 53.7 (d, J= 7,4Hz);” by “54.1 (d, J= 7.4Hz), 53.7 (d, J= 7.4Hz);”
Answer: We thank the Reviewer for that remark. This was corrected.
Page 7 Dibenzyl (1-hydroxyphenylmethyl)phosphonate (5d) : replace “5.09-4.83-(m, 5H),” by “5.09-4.83 (m, 5H),” and complete “68. (m);” 68.8 ?
Answer: We thank the Reviewer for that remark. This was corrected.
Page 7 Dimethyl 1-hydroxy-3-methyl-butylphosphonate (5e): replace “65.8 (d, J=160.6 Hz), 53.3-53.5(m), 39.9, 24.1 (d, J=14.3 Hz),” by “65.8 (d, J=160.6 Hz), 53.3-53.5(m), 39.9, 24.1 (d, J=14.3 Hz),”
Answer: We thank the Reviewer for that remark. This was corrected.
Page 7 Dibenzyl 1-hydroxy-3-methyl-butylphosphonate (5f) replace “68.1 (t, J=6.8 Hz),” by “68.1 (t, J=6.8 Hz),”
Answer: We thank the Reviewer for that remark. This was corrected.
Page 7 Dimethyl 1-hydroxy(pyridine-3-yl)methylphosphonate (5g): replace “δ 23.2 HRMS“ by ” δ 23.2; HRMS”
Answer: We thank the Reviewer for that remark. This was corrected.
Page 8 line 3 replace “Rection proceeded already at room temperature“ by “Reaction proceeded already at room temperature “
Answer: We thank the Reviewer for that remark. This was corrected
Page 8 lines 4, 5, 8, 19, 21, 23 and 24: replace “Table S1” by “Table SI1
Answer: We thank the Reviewer for that remark. This was corrected as requested
For the figure at the bottom of the page 10: could you please replace 16,21 by 16.21, 4,32 by 4.32, 14,77 by 14.77, 1,69 by 1.69, 7,22 by 7.22, 15,92 by 15.92, 3,72 by 3.72, 6,96 by 6.96, 14,46 by 14.46, 1,26 by 1.26, 5,23 by 5.23, 13,65 by 13.65, 2,94 by 2.94, 5,27 by 5.27, 15,42 by 15.42, 0,79 by 0.79, 3,45 by 3.45.
Answer: We thank the Reviewer for that remark. As the figure is generated from excel file, we are unable to change the comma for dot.
This paper is appropriate for publication in Molecules. The manuscript is well written in Standard American English. The referee recommends the publication of this article in Molecules after all the revisions are performed.
We would like to thank the reviewer for his kind remark about the pertinence of our study.
Many thanks for taking the time to review our changes; we hope these modifications are in accordance with the expectations of the reviewer.
Yours sincerely,
Claude Grison, on behalf of the co-authors

Reviewer 2 Report
The authors describe a green Eco-MgZnOx catalyst for the synthesis of α-hydroxyphosphonates via hydrophosphonylation of aldehydes under solventless conditions. The reactions gave excellent yields and the catalyst has wide-scope. The manuscript can be accepted subject to the following suggestions/comments:
Scheme 1. Has problems. Please fix.
Please comment of why Mg was lost from Eco-MgZnOx-F during recycling and remained intact when Eco-MgZnOx-F was used?
The authors should perhaps try to supplement the Eco-MgZnOx-F with added MgO to see if it indeed the cause of activity during the recycling of the catalyst.
Schemes 2 and 3: Add % to indicate the yield.
Scheme 2 and 3: The authors should try aldehydes with strong electron-donating groups like Meo. They also should try aldehydes with amino substituents to see if the amino group affects the reaction or not.
Line 51: for aliphatic ang heteroaromatic aldehydes should be for aliphatic and heteroaromatic aldehydes.
The figures and drawings are not clear. Please supply clearer and sharpers figures and drawings.
Author Response
Please find attached the new version of our manuscript, initially submitted under the manuscript Molecules article, for the special issue “Sustainable Chemistry in France”.
We would like to thank you for giving us the final opportunity to correct our manuscript. We believe that we have addressed the referee’s comments and have attached the necessary changes below.
Reviewer 2
The authors describe a green Eco-MgZnOx catalyst for the synthesis of α-hydroxyphosphonates via hydrophosphonylation of aldehydes under solventless conditions. The reactions gave excellent yields and the catalyst has wide-scope. The manuscript can be accepted subject to the following suggestions/comments:
Scheme 1. Has problems. Please fix.
Answer: Scheme 1 was replaced. We hope that now it is clearer now.
Please comment of why Mg was lost from Eco-MgZnOx-F during recycling and remained intact when Eco-MgZnOx-F was used?
Answer: We thank the Reviewer for that remark. In both cases (Eco-MgZnOx-P & Eco-MgZnOx-F), during work-up, a small quantity of ecocatalyst could not be recovered for the following reaction. Indeed, the MP-AES analyses of the mineral composition of the catalysts recovered after each reaction showed that the amount (wt%) of Zn, Mg and Ca is decreasing. This could be explained by the leaching of some ecocatalyst with the formed product, hydroxyphonates, known to be good chelating agents. The main difference between Eco-MgZnOx-P & Eco-MgZnOx-F stands in the initial concentration of Mg which is less than half in the latter ecocatalyst. The reactivity loss in the case of Eco-MgZnOx-F recycling is probably due to the fact that the minimum ecocatalyst threshold is reached, for the reaction to proceed fully, as early as the 2nd recycling step.
The authors should perhaps try to supplement the Eco-MgZnOx-F with added MgO to see if it indeed the cause of activity during the recycling of the catalyst.
Answer 2: We thank the Reviewer for that remark. As shown in Supporting Information, not only Mg is gradually leaching out together with the phosphonate product between recycling steps but also Zn, therefore a thorough study might be more complex than expected (this particular point could be investigated and improved in the future). In the case of Eco-MgZnOx-F, the minimum critical amount of ecocatalyst is reached more rapidly than for Eco-MgZnOx-P, as the initial concentration is lower. The role of MgO was shown rigorously using commercial mixtures, as summarized in Table 2, but it is also crucial to keep in mind that the ecocatalysts are not just metals added together, but instead complexe blends.
Schemes 2 and 3: Add % to indicate the yield.
Answer: Thank you for that remark however, we prefer to add % in the caption of the scheme 2 and 3. In our opinion it is better this way.
Scheme 2 and 3: The authors should try aldehydes with strong electron-donating groups like Meo. They also should try aldehydes with amino substituents to see if the amino group affects the reaction or not.
Answer: We thank the Reviewer for that remark however, we think that the presented substrate scope is already broad. Different electron withdrawing and electron donating substituents in the case of aldehydes (and also one unsaturated aldehyde) were used along with structurally different H-phosphonates. Including more examples could be considered but additional time would be necessary.
Line 51: for aliphatic ang heteroaromatic aldehydes should be for aliphatic and heteroaromatic aldehydes.
Answer: Thank you for that remark. This was corrected as requested.
The figures and drawings are not clear. Please supply clearer and sharpers figures and drawings.
Answer: Original high-resolution files in .tiff format were provided to the publisher separately with the corrected version of the manuscript.
We would like to thank the reviewer for his kind remark about the pertinence of our study.
Many thanks for taking the time to review our changes; we hope these modifications are in accordance with the expectations of the reviewer.
Yours sincerely,
Claude Grison, on behalf of the co-authors

Reviewer 3 Report
This manuscript reports application of plant-derived Mg- and Zn-containing catalysts in the hydrophosphorylation of aldehydes. These catalysts show good activity and good substrate scope. In general, this work remains a double impression. On the one hand, the production of cheap catalytic systems directly from biomass is an important trend in modern chemistry. But on the other hand, a number of cheap, very effective, and recyclable catalysts were already proposed for this reaction. Although the proposed catalysts provide good activity, their composition will be highly dependent on many factors, which is ultimately unacceptable for real application in a laboratory scale or in industry. Moreover, these ecocatalysts are unlikely to be economically advantageous over simple and very cheap compounds such as CaO, MgO, or Na2CO3. Meanwhile, I still believe that this work deserves publication in the specified special issue, and after addressing minor points listed below it can be accepted.
1. In Introduction, authors wrote: “…CaO,[46] MgO,[47] or Na2CO3,[48] were reported to promote the hydrophosphonylation of carbonyl substrates under solventless conditions. However, they were mostly used in stochiometric amounts…”. It is not always true. For instance, see Tetrahedron Letters 2021, 85, 153466.
2. In the scheme on Table 1, the H-phosphonate was displayed in unstable O-H form. Please note that the O-alkyl-substituted H-phosphonates exist in a P-H form, rather than O-H.
3. The picture in Figure 1 is of low resolution and poorly understandable for readers.
4. As seen from Table 1, both catalysts contain noticeable amounts of potassium and calcium, the percentage of which are comparable with those of Zn and Mg. However, the name of the ecocatalysts (“Eco-MgZnOx”) highlight only presence of Mg and Zn, but not reflects the presence of K and Ca, whose derivatives can also contribute to the catalytic activity. Please comment this issue.
Author Response
Please find attached the new version of our manuscript, initially submitted under the manuscript Molecules article, for the special issue “Sustainable Chemistry in France”.
We would like to thank you for giving us the final opportunity to correct our manuscript. We believe that we have addressed the referee’s comments and have attached the necessary changes below.
Reviewer 3
This manuscript reports application of plant-derived Mg- and Zn-containing catalysts in the hydrophosphorylation of aldehydes. These catalysts show good activity and good substrate scope. In general, this work remains a double impression. On the one hand, the production of cheap catalytic systems directly from biomass is an important trend in modern chemistry. But on the other hand, a number of cheap, very effective, and recyclable catalysts were already proposed for this reaction. Although the proposed catalysts provide good activity, their composition will be highly dependent on many factors, which is ultimately unacceptable for real application in a laboratory scale or in industry. Moreover, these ecocatalysts are unlikely to be economically advantageous over simple and very cheap compounds such as CaO, MgO, or Na2CO3. Meanwhile, I still believe that this work deserves publication in the specified special issue, and after addressing minor points listed below it can be accepted.
1. In Introduction, authors wrote: “…CaO,[46] MgO,[47] or Na2CO3,[48] were reported to promote the hydrophosphonylation of carbonyl substrates under solventless conditions. However, they were mostly used in stochiometric amounts…”. It is not always true. For instance, see Tetrahedron Letters 2021, 85, 153466.
Answer: Thank you for that remark. The reviewer is correct, it is rephrased as follow: However, they were often used in stochiometric amounts, the reactions were substrate specific and most importantly for green chemistry considerations, the production of the salts is far from environmentally friendly, with harsh production processes, poor impact assessment and positive carbon footprint (Büchel, K. H.; Moretto, H.-H.; Werner, D. In Industrial Inorganic Chemistry, 2nd Completely Revised Edition; 2000).
- In the scheme on Table 1, the H-phosphonate was displayed in unstable O-H form. Please note that the O-alkyl-substituted H-phosphonates exist in a P-H form, rather than O-H.
Answer: This was corrected as requested.
- The picture in Figure 1 is of low resolution and poorly understandable for readers.
Answer: Original high-resolution files in .tiff format were provided to the publisher separately with the corrected version of the manuscript to be
- As seen from Table 1, both catalysts contain noticeable amounts of potassium and calcium, the percentage of which are comparable with those of Zn and Mg. However, the name of the ecocatalysts (“Eco-MgZnOx”) highlight only presence of Mg and Zn, but not reflects the presence of K and Ca, whose derivatives can also contribute to the catalytic activity. Please comment this issue.
Answer: Thank you for that remark. In the preliminary experiments we showed that K2CO3 and CaCO3, possible species present in the ecocatalysts, are not catalysing the reaction well (61% and 0%, respectively). Although ZnO is also not good catalyst (only 17% conversion) we believe that Zn combined with Mg can have important role in the catalytic process. Namely, we postulate that the ecocatalysts act as plurifunctional catalyst where Mg species with basic character should be responsible for activation of the H-phosphonate and formation of a phosphanion (phosphite form). Whereas the presence of Zn species with Lewis acid character could be responsible for the activation of the carbonyl carbon making it more electrophilic and prone to reaction with phosphorus nucleophile. This assumption was also confirmed by the DFT studies described in the manuscript. Additionally, during the recyclability testing, MP-AES analyses of the mineral composition of the catalysts recovered after each reaction showed that while the amount (wt%) of Zn, Mg and Ca is decreasing, the amount of K remains constant (around 13.5 wt% and 11.5 wt% for Eco-MgZnOx-P & Eco-MgZnOx-F respectively). The drastic decrease in reactivity (14 % conversion) observed for Eco-MgZnOx-F for run 3, while the K concentration did not change tend to indicate that K is at least not the limiting catalytic species. For those reasons, we choose to name the catalysts Eco-MgZnOx
We would like to thank the reviewer for his kind remark about the pertinence of our study.
Many thanks for taking the time to review our changes; we hope these modifications are in accordance with the expectations of the reviewer.
Yours sincerely,
Claude Grison, on behalf of the co-authors
